# FlexLip: A Controllable Text-to-Lip System

**DOI:** 10.3390/s22114104

**Published:** 2022-05-28

**Authors:** Dan Oneață, Beáta Lőrincz, Adriana Stan, Horia Cucu

**Affiliations:** 1Speech and Dialogue Research Lab, University “Politehnica” of Bucharest, 060042 Bucharest, Romania; horia.cucu@upb.ro; 2Faculty of Mathematics and Computer Science, “Babeș-Bolyai” University, 400347 Cluj-Napoca, Romania; beata.lorincz@ubbcluj.ro; 3Department of Communications, Technical University of Cluj-Napoca, 400114 Cluj-Napoca, Romania; adriana.stan@com.utcluj.ro; 4Zevo Technology, 077042 Roșu, Chiajna, Romania

**Keywords:** text-to-lip, speech synthesis, text-to-speech, speech-to-lip, zero-shot adaptation, generative models, deep learning, artificial intelligence, objective measures

## Abstract

The task of converting text input into video content is becoming an important topic for synthetic media generation. Several methods have been proposed with some of them reaching close-to-natural performances in constrained tasks. In this paper, we tackle a subissue of the text-to-video generation problem, by converting the text into lip landmarks. However, we do this using a modular, controllable system architecture and evaluate each of its individual components. Our system, entitled FlexLip, is split into two separate modules: text-to-speech and speech-to-lip, both having underlying controllable deep neural network architectures. This modularity enables the easy replacement of each of its components, while also ensuring the fast adaptation to new speaker identities by disentangling or projecting the input features. We show that by using as little as 20 min of data for the audio generation component, and as little as 5 min for the speech-to-lip component, the objective measures of the generated lip landmarks are comparable with those obtained when using a larger set of training samples. We also introduce a series of objective evaluation measures over the complete flow of our system by taking into consideration several aspects of the data and system configuration. These aspects pertain to the quality and amount of training data, the use of pretrained models, and the data contained therein, as well as the identity of the target speaker; with regard to the latter, we show that we can perform zero-shot lip adaptation to an unseen identity by simply updating the shape of the lips in our model.

## 1. Introduction

Over the past few years, our society has constantly been increasing the amount of multimedia output. From online radio and television, to YouTube video bloggers and the popular Facebook Lives, professionals and nonexperts alike generate multimedia content at a tremendous pace. However, the costs and effort of generating high-quality multimedia content become increasingly significant, and many parties are already looking into deep learning for solutions to lessen the burden of pre- and postproduction, as well as end-to-end media generation. In the area of spoken content generation, text-to-speech (TTS) systems have already been, to a large extent, adopted by semiprofessional content creators, with the most-known platform for providing this service to its users being TikTok. However, when tackling the complete text-to-video synthesis, the solutions and quality of the available systems are not at the same level of integration into the media platforms. Even though there are numerous applications that it could address, such as anchor news delivery, video podcasts, gaming characters generation, and so on.

In this context, our work focuses on the task of rendering a video of a person delivering a spoken content starting from a given text, and optionally, a selected identity that can differ from the available training data. We decompose this complex task into a three module pipeline—(i) text-to-speech, (ii) speech-to-keypoints, and (iii) keypoints-to-video—and set out to derive a controllable and objectively measurable architecture for it. However, our work does not focus on the complete head movement and generation of facial characteristics, but rather it limits its scope to the generation of lip landmarks starting from an input text. As a result, we only tackle the first two modules of the complete pipeline described before and show an overview of our system in Figure 1. One reason for why we are not addressing the final module is the fact that up to this moment, there are no generally agreed upon objective measures for it, and most papers publish only perceptual, subjective evaluations.

Another important aspect of our work is the fact that most of the previous works address only the second module, i.e., speech-to-lips [1,2]. Having a controllable, easily adaptable TTS system integrated into the flow of the system can enable the end-user to control all aspects of the media generation system, including the spoken content and identity. As shown in Figure 1, the proposed architecture encompasses the ability to select various speech identities, alter the prosodic patterns, while also being able to control or disentangle the identity of the generated lips from the spoken one. For example, we could generate speech with the voice of former president Obam, using the lip shape of former president Trump and the face of former president Bush.

We can summarise the main contributions of our paper as follows:We propose a novel text-to-lips generation architecture, entitled FlexLip;We design its architecture as a flexible and highly controllable one;We analyse the effect of using synthesised speech as opposed to natural recordings;We propose a zero-shot adaptation of the speech-to-lips component;We show that by using as little as 20 min of spoken data, we can control the target speaker’s identity;We also show that the controllability of the architecture enables us to perform more accurate objective measures of its performance.

The paper is organised as follows: Section 2 introduces the works related to our proposed method, with the method being described in Section 3. The experimental setup and results are presented in Section 4 and Section 5, and their conclusions are drawn in Section 6.

## 2. Related Work

The task of generating video (i.e., a talking-head video) starting from speech or text has recently gained interest in the research community next to other tasks converting one modality to another, such as image-to-text or video-to-text, also called image or video captioning. For addressing the transformation of text into video output, various pipelines and different types of latent spaces were proposed. All studies that approach text-to-video conversion use an initial text-to-speech system to generate speech or speech features [3,4]. When going from speech to video, some studies argue that having an intermediate representation of the face or mouth can heighten the performance of the system [3,4,5], while in other cases, the authors approach the issue in an end-to-end manner, going from speech directly to video [1,2]. There are also studies that focus solely on the speech-to-keypoints task [6,7], as this is regarded as being more difficult than the subsequent keypoints-to-video task, as there is no direct, one-to-one mapping from the text input to the individual video frames. One of the first attempts to create a complete text-to-video pipeline was introduced in [8]. A simple idea was explored, whereby mouth images extracted from a video sequence were reordered to match a new, unseen phoneme sequence derived from the input text. In following works, text-to-video synthesis was approached as an extension of text-to-speech synthesis. Concatenative speech synthesis was extended to map characters to visemes, defined as static mouth shapes [9] or as temporal movements of the visual speech articulators [10]. Appropriate visemes were chosen from a large dataset for a particular speaker and morphed from one to the other in order to generate smooth transitions between successive visemes. In a more recent work, Fried et al. [11] approached text-to-video synthesis from a slightly different perspective by proposing a method to edit an existing video in order to reflect a new text input. This method still relies on a viseme search, concatenation, and blending. To generate mouth movements matching the edited text, visemes present in other parts of the video were used.

Inspired by the application of face keypoints in video prediction, where the keypoints guide the generation of future frames [12], most recent methods in text-to-video synthesis use face or mouth keypoints as intermediate representations [3,4]. Zhang et al. [4] addressed the task in two steps: (i) transforming the text into a sequence of keypoints (which the authors denote as poses), using a dictionary of (phoneme, keypoints) pairs; (ii) using a GAN-based architecture to generate video from interpolated phoneme poses. Simultaneously, the text was transformed into speech by a text-to-speech synthesis system. Kumar et al. [3] were the first to propose a sequence of fully trainable neural modules to address text-to-video conversion in three main steps. First, a text-to-speech system was used for audio generation starting from characters, not phonemes. A second network was then employed to generate mouth region keypoints synchronised with the synthetic speech. Finally, a video generation network produced video frames conditioned on the mouth region keypoints.

A couple of recent studies focused on parts of the complete text-to-video pipeline performing video synthesis directly from speech features [1,2,5] or approached just the speech-to-keypoints task [6,7]. In the video synthesis systems presented in [1,5], the mapping between audio features and mouth shapes are learnt by recurrent or convolutional neural networks. The audio input is paired with either mouth region keypoints [5] or with images of the target face [1] and the lip-synced video is predicted by the network. The approach for audio-to-video generation described in [2] is based on a neural network that includes a latent 3D representation of the face. As the keypoint-based intermediate representation seems to be a common choice in previous studies and also paves the way for creating speaker-independent systems, Eskimez et al. [6] proposed an LSTM trained with 27 subjects to solve the task that is able to generalise to new subjects. Greenwood et al. [7] generated full-face keypoints, as opposed to only mouth region keypoints, for two subjects using a BiLSTM.

For the evaluation of video synthesis methods, both objective and subjective measures are employed. In the objective evaluation, various difference metrics are computed between the real (ground-truth) and generated videos. Chen et al. [13] used the mean squared error, the Frechet video distance, based on the distance between features of the real and generated videos reported in [12]. Human evaluations are frequently used as subjective measures to capture the visual quality of the generated videos [12,14]. In a very recent paper, Aldeneh et al. [15] proposed a perceptual evaluation model that can be used to automatically and accurately estimate the subjective perceptual score for a given lip motion sequence.

With respect to text-to-speech synthesis (TTS), there are numerous neural architectures that achieve close to natural synthetic speech quality. Further, if at the beginning of the deep learning era for TTS the research was oriented towards full end-to-end systems going from input characters to audio waveforms [16], in recent years, the focus shifted towards flexible and controllable architectures [17,18,19]. This type of architecture enables several factors of the synthetic speech to be easily tuned during inference. This is the case for the FastPitch architecture [18], in which the duration, energy, and pitch of the output speech can be specified or copied from a different audio input. In the context of evaluating text-to-lip models, being able to control the duration of the synthesised speech means that the original phoneme durations of a natural speech sample can be replicated. This can provide a better alignment between the keypoints predicted from natural and synthetic input speech, and no substantial additional error is introduced by the alignment, making the objective evaluation more accurate.

In this context, our work resembles relatively well the work of Kumar et al. [3], but bears a few important distinctions. First and most importantly, our aim was to assess in a thorough and objective manner the quality of each component of a text-to-video pipeline. As such, we did not address the keypoints-to-video task, which is inherently a subjective one. Different from Kumar et al. [3], who do not evaluate their system subjectively, nor objectively, we carefully designed the text-to-keypoints pipeline to allow for objective evaluation and performed the evaluation of each component independently and at the system-level as well.

In terms of evaluation of generated lip movements, our work is, to the best of our knowledge, the first one to evaluate objectively the output of the text-to-keypoints task. Many works evaluate the quality of the generated sequence when the system is fed natural speech [6,7,15], but none of these start with text as input. One of the problems is that most of the neural-based TTS systems do not enable the exact control of the duration of the output; therefore, there is no one-to-one correspondence between the ground-truth frames and the synthesised ones. With respect to this, we believe that managing to objectively assess the quality for the text-to-keypoints seen as a whole is one of the important contributions of our work.

## 3. Text-to-Lip System Description

Our text-to-lip system is composed of two independent modules: **a text-to-speech synthesis system** and a **speech-to-lip** one. This independence ensures a more controllable setup and each module can be easily replaced. The following sections describe the two modules and their training procedures, while also focusing on their controllability.

### 3.1. The Text-to-Speech Component

When generating lip movements from speech, the quality of the input speech is essential. If the input contains natural speech recordings, they should also be high-quality. Therefore, for our text-to-speech synthesis component (TTS), we selected one of the latest deep-neural-based architectures, able to generate speech that is very close to the natural one. The architecture is FastPitch [18], and aside from its high performance, it uses a fast-inference parallel architecture, and enables the control of the pitch and duration of the input phonemes. The latter feature facilitates the tweaking of the output such that the speech-to-lip module is better fitted to the target speaker. FastPitch is based on bidirectional Transformers, which make up the encoder and decoder sections of the network. Separate paths are allocated for the pitch and duration prediction, as well as (if this is the case) for a speaker embedding layer. The encoder predicts one Mel-spectrogram frame per phoneme, which is then augmented with the pitch information, and upsampled according to the duration predictor. The prediction is then passed through the decoder to obtain the smoothed, complete Mel-spectrogram.

The Mel-spectrogram is then transformed into a waveform with the help of the WaveGlow neural vocoder [20]. WaveGlow uses a normalising flow-based architecture inspired from Glow [21] and WaveNet [16], but eliminates the autoregressive nature of them. The architecture uses a single network trained to maximise the likelihood of the data and, based on its flow nature, enables the computation of the true distribution of the training data.

As high-quality TTS systems commonly require large amounts of training data, we also adopt a fine-tuning procedure for the FastPitch model. Two pretrained models were used: a single speaker one, and a multispeaker one for which the network was not conditioned on the speaker identity. These models were then adapted to the target speaker using various amounts of speech data, as described in Section 5.

### 3.2. The Speech-to-Lips Component

This subsection describes the speech-to-lips component, which takes as input an audio of a person speaking and outputs the keypoints (the moving lips) that correspond to the spoken words. Since the task is a sequence-to-sequence one (we want to map a sequence of audio frames to a sequence of lip keypoints), we opt to implement the speech-to-lips module as a Transformer network, which has shown remarkable performance on many related tasks. The Transformer has two main components: an encoder module that uses self-attention layers to pool the input audio, and a decoder module that uses attention layers to aggregate information from both the encoded audio and previously generated lips. The decoder predicts the lips keypoints at each time step in an autoregressive manner, and is exposed to the entire input audio sequence.

**Transferring representations.** Given that the network processes audio streams, we decided to reuse the encoder architecture and its pretrained weights from a state-of-the-art speech recognition system. As such, we evaluate two variants of the network: one in which the encoder is frozen to the pretrained weights and we train only the decoder part; a second in which we train both components, the encoder and decoder. Note that training the decoder is mandatory because the speech recognition decoder is designed to output a sequence of characters, while in our task, the output is a sequence of keypoints.

**Lips keypoints preprocessing.** Video recordings of people talking involve variations in terms of their position, size, and head pose. Since this information affects the lips’ coordinates but is irrelevant to the task at hand, we remove these variations from our training data by transforming the absolute coordinates of the lips into a normalised space. More precisely, we apply the following three transformations to the extracted face landmarks: translate such that they are centred on the lips; rotate such that the line connecting the eyes is horizontal; scale such that the distance between the eyes is constant (we set an arbitrary value of five).

A second preprocessing step consists in projecting the normalised lip coordinates (40 coordinates of the lips: 20 keypoints with the *x* and *y* coordinates each) to a lower-dimensional manifold using principal component analysis (PCA); we denote the principal components by $v_{1},\ldots ,v_{D}$. We use an eight-dimensional space for projection (D=8), which captures around 97% of the variation of the training data. Figure 2 illustrates the axes of variation captured by the selected principal components. The reconstruction for component *i* and magnitude *s* is given by $\mu +\sum sv_{i}$, where μ is the mean lip shape and the scaling factor *s* ranges in {−1.5,1.1,…,1.5}. We observe that the principal components capture the following variations in the data: open versus closed mouth is modelled by the first and fifth components; 3D rotations (yaw, pitch, roll) are captured by the third (yaw), fourth (pitch), and sixth (roll) components; lip thickness varies across the second, fifth, seventh, and eight components.

To sum up, our method maps a stream of audio to a list of 8D points, which correspond to the PCA coefficients α. Note that both preprocessing steps are invertible; so, at test time, if we want to overlay the predicted lips on a given subject, we first reconstruct the lips l based on the predicted PCA coefficients
(1)$$l=\mu +\sum \alpha_{i}v_{i},$$
and then we reproject the normalised coordinates in the absolute coordinate space by inverting the scaling, rotation, and translation transforms.

**Zero-shot speaker adaptation.** When predicting the lip movements of an unseen speaker, we have observed that the lip dynamics are accurate, but unsurprisingly, their shape resembles the one of the trained subject. We propose a method to adapt the lip shape to the one of the new person by replacing the lips mean in the PCA reconstruction with the lips of the target speaker:(2)$$l^{\prime }=\mu^{\prime }+\sum \alpha_{i}v_{i},$$
where the coefficients α are obtained in the same way as before, but the mean $\mu^{\prime }$ is updated, and is computed from the target, unseen speaker. This operation is inexpensive (as the mean can be estimated on a few frames) and does not require retraining. Figure 3 shows the lip shapes of three speakers and how they vary along the first principal component. Even if the principal components were estimated on data from the first speaker, these qualitative results suggest that the variations obtained for new speakers are plausible.

## 4. Experimental Setup

This section presents the datasets used for training and evaluating the proposed methods (Section 4.1) and implementation details regarding the two components of our system (Section 4.2).

### 4.1. Datasets

In order to evaluate a text-to-lip system, we need a three-modality dataset: video, audio, and text. However, there are very few large, freely available datasets that accomplish this requirement. In this context, we decided to build our own dataset comprising high-quality data available on YouTube. For the audio part, we also relied on well-established speech datasets. All the datasets used in our experiments are described in the following subsections.

#### 4.1.1. Obama Dataset

The former president of the United States, Barack Obama, has many videos on YouTube in which he addresses the nation in a systematic fashion: front-facing the camera and speaking clearly, with most of the videos being recorded in his office. From this series of weekly addresses, we downloaded a set of 301 videos, which were originally introduced by Suwajanakorn et al. [5] and subsequently used by Kumar et al. [3]. The YouTube videos come along with both manual and automatically generated closed captions. While the automatic captions are better aligned to the speech than the manual captions, the latter are more accurate and also include punctuation. The punctuation is essential for splitting the audio into sentence-length chunks, required by the TTS system. We split the audio into sentences based on the end-of-sentence punctuation marks encountered in the manual captions. This procedure leads to a set of around 10 k audio–video chunks and their approximate transcripts. The duration of the chunks is between 1 and 20 s long with transcripts between 15 to 500 characters. Very short and very long utterances were discarded, as the TTS architecture has problems attending to short sequences, as well audio sequences longer than 30 s. Some text examples are listed below:


*Under my Administration, we’re producing more oil than at any other time in the last eight years.*



*It had the support of 52 Democrats and 22 Republicans.*



*It’s our job as citizens to make sure we keep pushing this country we love toward our most cherished ideals—that all of us are created equal, and all of us deserve an equal shot.*


The total duration of the data is around 17 h, with most of the videos having a resolution of 720p and a frame rate of 30 frames per second. The audio is sampled at 44.1 kHz and has a bit depth of 16. To speed up the preprocessing (in particular the face landmark extraction), we downscaled the videos to 360p; for the speech-to-lip and TTS systems training, we downsampled the audio data to 22 kHz and maintain the 16 bps. Although the videos are recorded in quiet conditions, some reverberation and background noise are present; therefore, we preprocessed the audio through the Postfish tool using the default parameters (https://github.com/ePirat/Postfish (accessed on 15 May 2022)). Volume normalisation and silence trimming were performed using the Librosa tool (https://librosa.org/doc/latest/index.html (accessed on 15 May 2022)).

The dataset was then split into train–validation–test subsets at video-level, meaning there are no samples from a video that are independently assigned to the different splits. The test set was composed of 500 utterance-length samples, randomly selected from 30 videos, and was manually checked for alignment and transcription errors.

**Text processing and data selection.** Although the manual transcripts seemed to be of very high-quality, combined with the splitting algorithm, we noticed that the correspondence between the audio and the text was not completely accurate. Therefore, we performed a series of postprocessing steps. The first step included the normalisation of all nonalphabetic symbols present in the captions, such as numbers, currency symbols, and so on. A first pass was performed using the Num2Words Python package (https://pypi.org/project/num2words/ (accessed on 15 May 2022)); then, the entire dataset was manually checked for nonalphabetic symbols. The second step involved the use of an automatic speech recognition (ASR) system [22] to transcribe the audio chunks. A word error rate (WER) measure was computed between the ASR results and the closed captions. High WER utterances were discarded. Even with a highly accurate ASR system, errors can still propagate to the transcripts. As a result, we ran an additional quality check based on the TTS system’s loss. We used an intermediate checkpoint trained on the entire Obama data, set the batch size to 1, froze the layers, and ran the low WER utterances through the architecture. Audio chunks exhibiting a low loss measure were considered as having the correct transcripts, as the TTS system’s loss function includes the alignment of audio-to-text. From this step, a set of 3281 samples were retained. In order to extend the dataset as much as possible, we also manually checked 1480 samples and added them to the training set. The total duration of the final audio dataset is 8 h.

Due to the rather complex grapheme-to-phoneme rules present in English, which can make it hard for a TTS system to learn the appropriate alignment and pronunciation, we performed the phonetic transcription of the text prompts associated with the audio data. The front-end tool of the Festival TTS system [23] was used to generate the phonetic representations of the transcripts.

#### 4.1.2. Trump Dataset

To test the capabilities of our speech-to-lip system to adapt to unseen speakers, we collected a small dataset of a different speaker: the former president, Donald Trump. We manually inspected multiple videos of him found on YouTube and finally selected two that satisfy the following criteria: are reasonably long; he is the single speaker appearing; his head is mostly forward-facing, without extreme pose variations. The first video (https://www.youtube.com/watch?v=KJTlo4bQL5c (accessed on 15 May 2022)) is around one hour long and represents his speech at the Conservative Political Action Conference; this video was used for training. The second video (https://www.youtube.com/watch?v=xrPZBTNjX_o (accessed on 15 May 2022)) is almost ten minutes long and represents his presidential address on the COVID-19 pandemic; this video was used for testing. As for the Obama dataset, we downscaled the video to a resolution of 360p. As we did not use this speaker for the text-to-speech component, we did not extract the textual component of the data.

#### 4.1.3. Datasets for TTS and ASR Models’ Pretraining

Both text-to-speech and speech-to-lip components use pretrained models and fine-tuning towards the target speaker. The following datasets were used for the model pretraining.

LibriSpeech is a corpus of approximately 1000 h of 16-kHz, multispeaker, read English speech derived from read audiobooks of the LibriVox project, and has been carefully segmented and aligned [24]. LibriSpeech was used in our experiments to pretrain the ASR-based feature extractor in the speech-to-lip module.

LibriTTS [25] is derived from the LibriSpeech corpus and includes multispeaker English data of approximately 585 h sampled at a 24-kHz sampling rate. The LibriTTS corpus is designed for TTS research and has normalised text transcripts, as well as an automatic selection of samples that do not contain severe background noise and reverberation. LJSpeech dataset [26] is a high-quality single female speaker dataset containing around 24 h of read speech audio clips. The data are transcribed and aligned at utterance level. LibriTTS and LJSpeech were used to pretrain the TTS models.

### 4.2. Implementation Details

**Text-to-speech.** For the TTS models’ training process, we focused on the Obama dataset. In our initial experiments, we found that training a system only on the Obama data yields subpar results in terms of naturalness of the synthetic output (see Table 1, system id O-8). Therefore, we decided to pretrain two large models on the LJSpeech and LibriTTS datasets, and to fine-tune them towards our target data. The models were trained over 1000 epochs and use the phonetic transcription provided by the Festival tool. In the LibriTTS model, although it contains multiple speakers, we did not condition the output on the speaker id, but rather aimed at obtaining an eigen voicelike model. For this model, as the data are recorded in semiprofessional environments, we also performed a dereverberation preprocessing step. We can summarise the two pretrained models as follows:**LJ-24h**—trained on the 24 h of LJSpeech dataset recordings, which translate into 13,077 utterances;**LT-47h**—trained on a subset of 31,526 utterances (around 47 h belonging to 558 speakers) from the LibriTTS dataset.

Starting from the phonetic transcription, dereverberation process, and pretrained models, we carried on the training procedure for several TTS systems using the Obama dataset. The systems pertain to the different pretrained models, the use of dereverberation for the Obama samples, and the amount of data selected to perform fine-tuning. The list of Obama TTS systems is presented in Table 1, along with their objective evaluation in terms of word error rate (WER) and cosine similarity to the natural recordings. The fine-tuning was run over the data for an additional 500 epochs.

**Speech-to-lip.** The input to the speech-to-lip network are Mel filterbank features extracted from the audio files. The output consists of lips landmarks extracted automatically from the corresponding video. Concretely, we used the dlib library [27] to extract facial landmarks, which produced 68 landmarks for each frame; we keep those twenty landmarks corresponding to the lips (indices 48–68). Occasionally, dlib detects zero or more than one face (most of the time erroneously since there is a single face appearing in the video shots); we deal with these exceptions as follows: if there is no face detected, we interpolate in time based on the neighbouring faces; if there is more than one face detected, we keep the one with the largest confidence. Next, we project the absolute coordinates into a normalised space by translating, rotating, and scaling (translate to centre the lips, rotate to have the eye-line horizontal, scale to ensure that the distance between eyes is five pixels). Finally, the lips are projected to a low-dimensional space using a principal component analysis (PCA) model fitted on a subset of landmarks from the train split of the Obama dataset. We use the top eight principal components, which capture about 97% of the total data variation.

When training the speech-to-lip network, in order to have balanced batches and improve the speed, each video shot is split into eleven-second chunks with an overlap of one second. At test time, if we are given an audio file longer than eleven seconds, we carry out a similar procedure as for training: we split the audio file into eleven-second chunks with a second overlap; predict the lips for each of the chunks separately; then, average out the overlapping regions (the averaging operation is carried in the low-dimensional space of the PCA).

The training is set to 100 epochs and the final model averages the weights of the top ten best models on the validation set [28]. We use a warm-up learning scheduler, which increases the learning rate linearly for 5000 steps up to 0.02 and then decreases it as a function of $1/\sqrt{s}$, where *s* is the step value. The batch size is set to 6 samples.

## 5. Experimental Results

This section presents quantitative results for the two components described in the paper (the text-to-speech module, in Section 5.1, and the speech-to-lips module, in Section 5.2) as well as their integration (in Section 5.3). We also provide qualitative samples corresponding to these results at the following link: https://zevo-tech.com/humans/flexlip/ (accessed on 15 May 2022).

### 5.1. Objective Evaluation of the TTS Models

The quality of a TTS system in general pertains to its naturalness and intelligibility, as well as speaker similarity. To evaluate the TTS systems, we analysed the objective measures of word error rate (WER) and a speaker–encoder-based cosine similarity over the entire Obama test set. These two measures have recently been found to have a high correlation to the perceptual measures obtained in listening tests [29,30]. The WERs are extracted based on the automatic transcripts provided by the SpeechBrain ASR system [22], while the cosine similarity uses SpeechBrain’s speaker embedding network. Both the ASR and the speaker embedding networks are based on an ECAPA-TDNN architecture [31]. For the speaker similarity measure, we averaged the cosine similarity measure values between the synthesised samples and their natural counterparts. To estimate the cosine similarity over the natural samples, we employed an intradata evaluation and compared random pairs of samples. This means that the linguistic content is different, and it can affect the estimation of the speaker similarity, as these types of speaker embedding networks do not truly factor out the spoken content and background conditions. The results are shown in Table 1. We explore several different dimensions of the TTS system’s training procedure: (i) using only the target speaker’s data (id: O-8) vs. using a pretrained model (ids: LT-* and LJ-*); (ii) using dereverberation over the target data; (iii) number of training samples used from the target speaker. We should point out the fact that the 1- and 0.3-hour subsets are selected from the manually checked training set, and that the dereverberation refers to the target speaker’s data, not the pretraining data. By inspecting the results, the first interesting thing to notice is that the WER over the natural samples (id: Natural) is worse than the best-performing system (id: LJ-8). However, during the manual check of the test set, we noticed that some of the samples contained high background noise and reverberation. This can definitely affect the performance of the ASR system. Although, even after applying the dereverberation method over the natural samples (id: Natural-dvb), the results were similar. We could explain this by the fact that the background noise is one of the major causes of degradation, while for the TTS system, the different background conditions are, in principle, averaged out within the model.

**Dereverberation.** Being also within the area of background conditions, the dereverberation algorithm was employed as a measure to improve the quality of the output synthetic speech; the cosine similarity measures support this preprocessing step. All systems trained with the dereverberated data exhibit higher cosine similarity measures with the natural samples. On the other side, in terms of WER, it seems that only when using the manually checked speech data does the dereverberation improves the overall results. When using all the data available (ids: LJ-8 and LT-8), the results over the dereverberated data are not as good as for the original dataset. The interpretation of these results can be based on the fact that, although the full target speaker dataset may still contain some transcription errors and various background noises, the high amount of available data is able to leverage the errors. It is also true that, although the dereverberation algorithm ensures a better perceptual quality of the audio samples, it may introduce signal-level artefacts that will interfere with the model training step. When using a smaller amount of the target speaker’s data (i.e., 20 min or 1 h), any improvement of the training data is directly transposed into the output of the synthesis system, and the dereverberated versions of these systems perform better than their non-dereverberated counterparts.

**Pretrained models.** With respect to the use of pretrained models, when using only the target speaker’s data (id: O-8), the WER and cosine similarity results are less-performing than any of the fine-tuned models. It also appears that, even though the complete 8-h dataset from the target speaker may still contain alignment errors between the audio and the transcript, these are not reflected in the overall results of the LJ-8 and LT-8 systems. This concludes the fact that having large amounts of data can average out some of the errors in the transcript. However, using only an eighth of the speech data (i.e., 1 h) can nearly match the top-line results of our TTS systems (see system id LT-1). Another result of our analysis is the fact that having multiple speakers in the pretrained model can provide a better starting point for our target speaker adaptation—comparing LJ-* with LT-* systems in terms of WER—and improve its intelligibility. However, it does not influence the speaker similarity, where the results are rather similar.

**Amount of training data.** As a general conclusion, using as little as 20 min of transcribed data can achieve similar results as the top-line systems. This means that in scenarios where only limited data are available, given a pretrained multispeaker model, good-quality TTS systems for the target speaker can still be obtained.

### 5.2. Evaluating the Speech-to-Lips System

This section presents an empirical evaluation of the speech-to-lips networks and their variants. We measure the performance of the systems by mean squared error (MSE) between the ground-truth lips and the predicted lips averaged across the number of keypoints, frames, and video segments. Unless specified otherwise, the MSE is computed in the normalised and low-dimensional space (eight-dimensional).

**Transferring representations.** Instead of training the speech-to-lip network from scratch (from random initial weights), we incorporate learned audio representations into the speech-to-lip module by transfer learning. We initialise the audio encoder from a state-of-the-art ASR system and evaluate two variants: either keep the encoder frozen or allow to update its weights together with the decoder’s. The results are presented in Table 2. We observe that the best results are obtained when the audio encoder is initialised from the ASR and is fine-tuned with the rest of the system. This approach has conceptual advantages over the other two variants: compared with a fully random initialisation, it has the benefit of reusing learnt information; compared with the frozen pretrained encoder, it has the advantage of being more flexible.

**Speaker adaptation and zero-shot adaptation.** In the next set of experiments, we investigate the best ways of reusing a pretrained speech-to-lip model for an unseen speaker. We explore three adaptation strategies: (i) Applying the pretrained model “as is” on audio data from the new speaker; (ii) Fine-tuning the pretrained model on a small amount of data from the new speaker (we attempt with datasets of 5, 10, 20, and 40 min); (iii) performing zero-shot adaptation by updating the PCA mean with the lip shape of the unseen speaker (as described in Section 3.2).

**The impact of data.** To understand how the amount of training data affects the speech-to-lip model, we performed a systematic study in which we trained the model on varying quantities of data: from the full training dataset (of 16 h) to smaller fractions (eight hours, four hours, and so on until only 15 min). The subsampling of the training data was carried at the video level, and not at the chunk level. We believe that this setup is more realistic, as it emulates the scenario in which we have access only to a few videos. Note, however, that this subsampling strategy might yield less-diverse training samples than subsampling chunks from the entire training dataset. For these experiments, we adjusted the warm-up cycle of the learning rate for the smaller datasets. More precisely, we linearly scaled the number of warm-up steps with the fraction of the data used.

The quantitative results are presented in Figure 4. We notice two regimes: (i) using at least four hours of video seems to yield reasonable performance, close to that obtained using the entire dataset of sixteen hours; (ii) using less than one hour of data, the results are degraded.

We start from the best speech-to-lip model trained on the full Obama data, with the encoder initialised from an ASR and fully fine-tuned (the model corresponding to the last row in Table 2). The evaluation setup follows the previous one, but differs in a couple of aspects:Data: for evaluation and fine-tuning, we used the Trump dataset, which was trawled from the internet, as described in Section 4.1.2. For simplicity, we evaluate on chunks of data (eleven-second chunks with overlap of one second), preprocessed in the same way as for training; based on the experiments on the Obama dataset, we have found very similar results for the per-chunk and per-sentence evaluations.Evaluation metrics: in addition to the mean squared error (MSE) computed in the normalised PCA space (8-dimensional; 8D), we also report the MSE computed in the original, 40-dimensional (40D) space. This second evaluation is especially relevant for the zero-shot adaptation method, which only affects the reconstructed lips.

The results are presented in Table 3. We observe that the first adaptation approach, which directly uses the Obama-trained model, yields a performance of 0.345 MSE 8D or 0.071 MSE 40D. Unsurprisingly, the results are much worse than what we observed when applying the model to the in-domain Obama dataset, 0.064 MSE 8D (as shown in Table 2), likely due to the mismatch of the datasets. The performance can be improved by the second adaptation method, fine-tuning on the speaker-specific data, which leads to better results with more data, reaching the best value of 0.096 MSE 8D and 0.021 MSE 40D (shown in bold in Table 3). These results are close to those obtained on the Obama dataset with around two–four hours of data (see Figure 4), showing that starting from a pretrained model can alleviate the need of large quantities of training data. Finally, we see that the proposed zero-shot adaptation method yields a significant improvement over the baseline: the error halves from 0.071 to 0.034 MSE 40D (the figure in italics from Table 3), which is close to the best result we achieve, of 0.021 MSE 40D. Note that this performance improvement is obtained without performing any training on the new speaker: we just update the PCA mean at test time.

As an additional experiment, we attempt to combine the second and third approaches for adaptation: we update the mean of the PCA to Trump’s lips shape also for the fine-tuned models. While the results are better than the zero-shot variant, they do not improve over the fine-tune-only variant that uses Obama’s lips shape. We believe that this happens because the fine-tuning process helps in adjusting for the new speaker’s lip shape, which then—when changed at test time—negatively affects the system.

### 5.3. Evaluating the Complete Text-to-Keypoints System

In this section, we evaluate our text-to-lip method in an end-to-end manner—that is, given a text input, we want to asses the quality of the generated lips. A major challenge of this joint evaluation is that the generated lips are not guaranteed to be synchronised with the ground-truth lips, since the synthesised audio is not necessarily synchronised with the natural audio. To address this issue, we propose two approaches, both aimed at aligning the intermediary audio representation. The first approach uses dynamic time warping (DTW) to align the Mel-frequency cepstral coefficient (MFCC) representation of the two audio sources. We use forty-dimensional MFCCs, and to facilitate the transfer of the alignment at the lip level, we extract the MFCCs using a hop length that yields the same number of coefficient vectors as the number of video frames. In the second approach, we make crucial use of our model’s ability to control the phoneme-level durations within the TTS system. More precisely, we set the durations to those obtained by running a forced-aligner over the phonetically transcribed evaluation text and its corresponding natural audio.

For the current evaluation, we consider the best TTS models as determined from the objective evaluation, i.e., LJ-8 and LT-8 (see Table 1) in terms of WER. We consider that the intelligibility of the speech is more likely to affect the correct lip movement, as opposed to having a speech input, which has a smaller speaker similarity measure. To estimate an upper bound of the performance, we also measure the performance obtained by starting with natural audio. As in the previous experiments, we report the mean squared error (MSE) computed in the 8D PCA space between the generated lips and the automatically extracted landmarks from the original video sequence.

The results are shown in Table 4. We can observe that using a DTW-based alignment, the estimated MSE measure is worse than the one obtained when using the original phone durations. This means that being able to control this particular aspect of the generated audio enables us to perform a more accurate evaluation of the speech-to-lip component. With respect to the natural vs. synthesised speech, the differences are not substantial, 0.094 vs. 0.064 in favour of natural speech. These differences partially pertain to the fact that the forced aligner is not perfect, and slight alignment errors are still present between the natural and synthesised audios. However, it is impossible to evaluate how much of the total error is determined by the misalignments versus the lip landmark generation network. Perceptual differences over the lip generation performance between the two types of speech inputs can also be analysed from our samples’ page (https://zevo-tech.com/humans/flexlip/ (accessed on 15 May 2022)).

## 6. Conclusions

In this paper, we proposed a flexible text-to-speech-to-lip pipeline that allows the user to control various facets of its outputs: the phone durations, the pitch contour of the voice, the audio altogether, the shape of the lips. Our model is based on two components: a text-to-speech network and a speech-to-lip network. For the text-to-speech component, we proposed using the FastPitch [18] architecture, which we carefully evaluated in multiple settings (involving the quantity and quality of the training data) and showed that the synthetic speech of the best models is intelligible and resembles the original speaker’s voice. For the speech-to-lip part, we used a Transformer network to map the Mel-spectrogram representation of the audio to PCA-encoded lips, which capture the dynamics of the lip movements. We have made the observation that the shape is encoded by the PCA mean and this can be easily replaced at test time (in a zero-shot adaptation setting), yielding results close to those obtained when retraining with 40 min of data. Finally, we have made crucial use of the controllability of our pipeline to carry out an objective end-to-end evaluation—by setting the phone durations to match the natural audio, we managed to obtain synchronised generated and natural lips, ensuring a relevant score. Importantly, these results have shown that the speech-to-lip module is robust to using synthetic data at input, as the performance of the full pipeline is close to that of the speech-to-lip with natural audio. As future work, we plan to extend our pipeline to the video domain (with a keypoints-to-video component) and use the proposed objective evaluation approach to evaluate objectively the output of the end-to-end system.

## Figures and Tables

**Figure 1 sensors-22-04104-f001:**
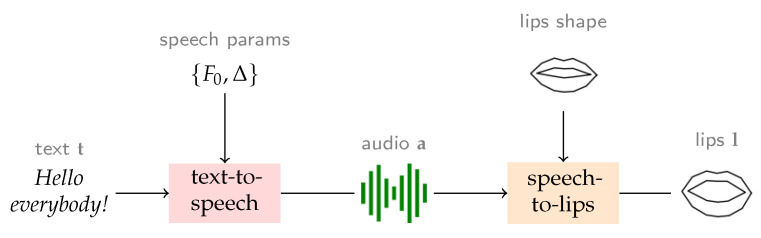
Schematic illustration of the proposed FlexLip pipeline. Our approach allows for a high degree of controllability by explicitly passing through the audio modality and permitting to specify speech parameters (fundamental frequency $F_{0}$, and phoneme durations Δ) and lips shape (as the mean shape upon which the learnt displacements are applied).

**Figure 2 sensors-22-04104-f002:**
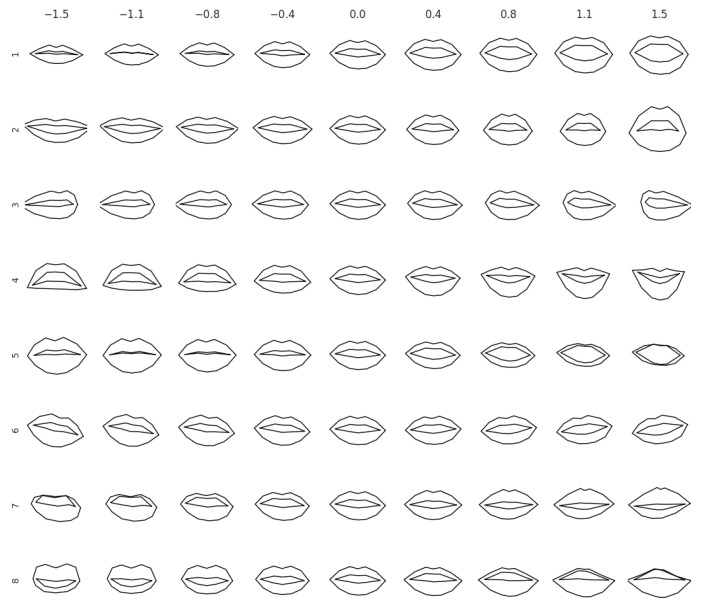
Axes of lips variation. On each row, we show the variation of the lips captured by one of the top eight principal components. The reconstructions are obtained by adding the scaled principal component to the mean lip shape. The scaling factor ranges from −1.5 to 1.5, as indicated on the top of the columns.

**Figure 3 sensors-22-04104-f003:**
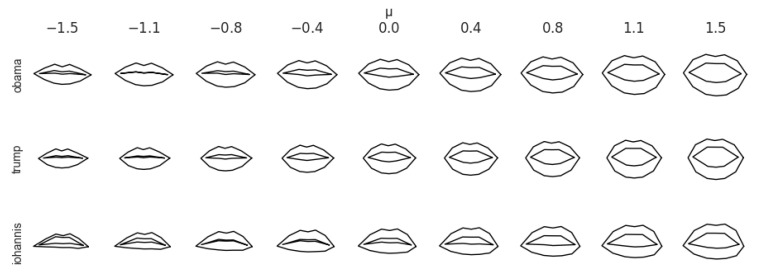
Lip shapes. The middle column, denoted by μ, represents the shapes of the lips for three speakers (one along each row). The columns show the variation of the shapes along the first principal component v; more precisely, the lips in row *r* and column *c* are computed as $\mu_{r}+\sum s_{c}v$, with $\mu_{r}$ being the (mean) lip shape and $s_{c}$ a scaling factor that ranges in $\left\{-1.5,1.1,\ldots ,1.5\right\}$. The PCA was fitted on data from the first speaker.

**Figure 4 sensors-22-04104-f004:**
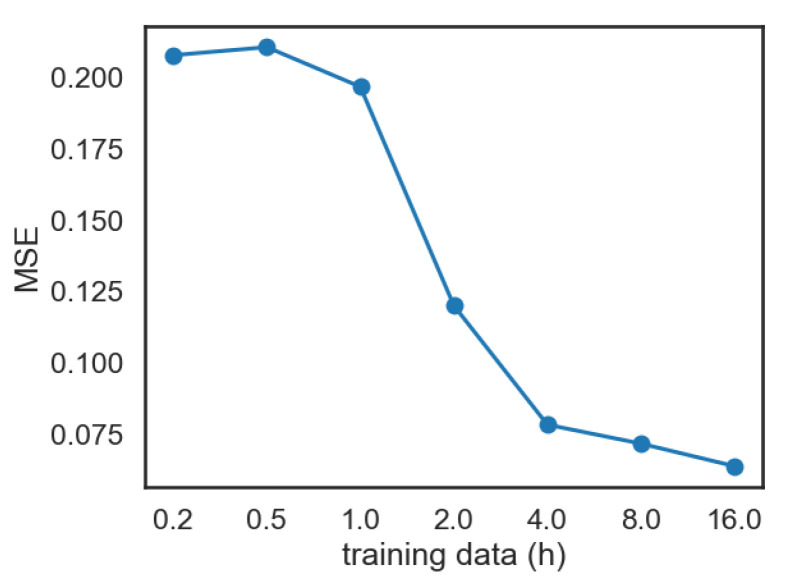
The impact of training data. We report the mean squared error (MSE) on the Obama test set for multiple speech-to-lip networks trained on varying fractions of data—1/64, 1/32, 1/16, and so on—up to the entire dataset, which consists of around 16 h of data. All models have their encoder initialised from a pretrained ASR and fully trained (not frozen).

**Table 1 sensors-22-04104-t001:** TTS models trained for the target speaker and the objective evaluation over the test set in terms of WER and cosine similarity to the natural recordings. The dereverberation column refers to the target speaker’s data. The cosine similarity for the Natural and Natural-dvb rows is computed as an intradata measure by comparing random sample pairs from within the speech set. The arrows in the table header indicate the direction of best performance for the respective measure. Boldface numbers highlight the best results for the respective column.

	Training Data					Cosine
**ID**	**Duration (h)**	**No. of Utts**	**Dereverb**	**Init.**	**WER**↓	**Similarity**↑
Natural	8	4761	no	–	9.32	0.684
Natural-dvb	8	4761	yes	–	9.22	0.683
O-8	8	4761	no	–	14.72	0.673
LJ-8	8	4761	no	LJ-24h	8.48	0.697
LT-8	8	4761	no	LT-47h	**7.31**	0.709
LJ-8-dvb	8	4761	yes	LJ-24h	11.45	**0.723**
LT-8-dvb	8	4761	yes	LT-47h	11.42	0.683
LJ-1	1	545	no	LJ-24h	10.50	0.690
LT-1	1	545	no	LT-47h	8.76	0.690
LJ-1-dvb	1	545	yes	LJ-24h	9.48	0.722
LT-1-dvb	1	545	yes	LT-47h	8.64	0.713
LJ-0.3	0.3	175	no	LJ-24h	12.98	0.679
LT-0.3	0.3	175	no	LT-47h	9.68	0.677
LJ-0.3-dvb	0.3	175	yes	LJ-24h	11.43	0.704
LT-0.3-dvb	0.3	175	yes	LT-47h	9.08	0.681

**Table 2 sensors-22-04104-t002:** Transferring representations. Evaluation of the speech-to-lip system in terms of the mean squared error (MSE) on the Obama test set. We consider three combinations of encoding the audio input in terms of initialisation (either random or from a pretrained automatic speech recognition system) and whether we train this component or keep it frozen. The arrows in the table header indicate the direction of best performance. Boldface numbers highlight the best results for the respective column.

Encoder		
**Init.**	**Train**	**MSE**↓
random	yes	0.130
from ASR	frozen	0.132
from ASR	yes	**0.064**

**Table 3 sensors-22-04104-t003:** Speaker adaptation and zero-shot adaptation. We report two variants of the mean squared error (MSE) on the Trump test set: one computed in the 8-dimensional PCA space (MSE 8D), the other computed in the reconstructed 40-dimensional original space (MSE 40D). The table presents results for three adaptation methods, as follows: 1. The first two entries from the first row correspond to the system pretrained on the Obama dataset. 2. The entries in rows 2–5 correspond to the pretrained model fine-tuned on various amounts of Trump data. 3. The third column corresponds to updating the PCA mean at test time to Trump’s shape; the first entry in the third column (shown in italics) represents the proposed zero-shot speaker adaptation. The arrows in the table header represent the direction of the best performance for the respective measure. Boldface numbers highlight the best results for the respective column.

		MSE ↓ 8D	MSE ↓ 40D	
	**Training**		**PCA Mean**	
	**Data**		**Obama**	**Trump**
1	obama	0.345	0.071	*0.034*
2	trump 5 m	0.109	0.024	**0.024**
3	trump 10 m	0.101	0.022	0.032
4	trump 20 m	0.099	0.022	0.031
5	trump 40 m	**0.096**	**0.021**	0.030

**Table 4 sensors-22-04104-t004:** From text to keypoints. We report the mean squared error on the Obama test set with respect to the use of natural vs. synthetic audio data. We also report the MSE values over the Dynamic Time Warping (DTW) alignments of the synthetic audio to the natural audio vs. controlling the phoneme durations (phone Δ) within the TTS system. The arrow in the table header represent the direction of best performance for the respective measure.

Audio	Alignment	MSE ↓
Natural	–	0.064
TTS · LJ-8	DTW	0.179
TTS · LT-8	DTW	0.181
TTS · LJ-8	phone Δ	0.095
TTS · LT-8	phone Δ	0.094

## Data Availability

The processed datasets used in this paper can be obtained from the authors.

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
