# Peer review of "FlexLip: A Controllable Text-to-Lip System"

_sensors, 2022, doi:10.3390/s22114104_

Round 1

Reviewer 1 Report

This paper tackles a sub-issue of the text-to-video generation problem, by converting the text into lip landmarks. This system, entitled FlexLip, is split into two separate modules: text-to-speech and speech-to-lip, both having underlying controllable deep neural network architectures. This modularity enables the easy replacement of each of its components.

While this issue need to be clarified,

1.  Line 256,’ Very short and very long utterances were discarded’, please state the reason.

2.  Experiment is conducted by comparing the parameter setting of your own approach. Why not compare with other text-to-lip approaches?

Author Response

26th of May, 2022

Response letter for the review of MDPI sensors-1750936

Dan Oneață, Beáta LÅ‘rincz, Adriana Stan, Horia Cucu,

”FlexLip: A controllable text-to-lip system”

We would like to first thank our editor and reviewers for the time and effort they have put in to evaluate our work, and for the extremely useful comments and suggestions they made in their reviews. 

We attempted to address all their points and listed below our responses, and the changes made to the article:

Reviewer 1:

  1. Line 256,’ Very short and very long utterances were discarded’, please state the reason.

We added an explanation in the paper:

”Very short and very long utterances were discarded, as the TTS architecture has problems attending to short sequences, as well audio sequences longer than 30 seconds.”

  1. Experiment is conducted by comparing the parameter setting of your own approach. Why not compare with other text-to-lip approaches?

There is little prior work on the full text-to-lip task. The work of Kumar et al. [3] is one such example, but unfortunately their paper does not provide any (quantitative) results and it does not point to an official implementation. While unofficial repositories do exist [a, b], they are incomplete, missing the text-to-speech component. In terms of the speech-to-lip component, it is difficult to compare directly to previous works since these evaluate on different datasets [4, 6, 7, 15] or using different metrics or splits [5]; we have also had difficulties in running the available codebases for some of these methods (for example, the code [c] corresponding to [5] provides only a partial implementation of the method discussed in the paper).

Best regards,           

Dan Oneață,             

Beáta LÅ‘rincz,             

Adriana Stan,             

Horia Cucu

Reviewer 2 Report

This is a very good paper. I have some minor comments as follows.

(1) A visual representation of the evidence of "We observe that the principal components capture the following variations in the data: open versus closed mouth is modelled by the first and fifth components; 3D rotations (yaw, pitch, roll) are captured by the third (yaw), fourth (pitch) and sixth (roll) components; lip thickness varies across the second, fifth, seventh and eight components. " would be great.

(2) What is the dimension of MFCC you used?

(3) Can you provide some comparative results?

Author Response

26th of May, 2022

Response letter for the review of MDPI sensors-1750936

Dan Oneață, Beáta LÅ‘rincz, Adriana Stan, Horia Cucu,

”FlexLip: A controllable text-to-lip system”

We would like to first thank our editor and reviewers for the time and effort they have put in to evaluate our work, and for the extremely useful comments and suggestions they made in their reviews. 

We attempted to address all their points and listed below our responses, and the changes made to the article:

Reviewer 2:

(1) A visual representation of the evidence of "We observe that the principal components capture the following variations in the data: open versus closed mouth is modelled by the first and fifth components; 3D rotations (yaw, pitch, roll) are captured by the third (yaw), fourth (pitch) and sixth (roll) components; lip thickness varies across the second, fifth, seventh and eight components. " would be great.

These observations can be derived from Figure 2, where each row corresponds to one of the variations described in the text. For example, the third row, which corresponds to the third principal component, shows rotations along the vertical axis (the yaw axis)—the leftmost image shows the lips rotated to the left, the rightmost image shows the lips rotated to the right. Similar conclusions can be drawn for the other axes of variations: open–closed mouth (first and fifth rows), other 3D rotations (fourth and sixth rows), lip thickness (second, fifth, seventh and eighth rows).

(2) What is the dimension of MFCC you used?

We have modified the paper to include this additional information (highlighted below):

“The first approach uses dynamic time warping (DTW) to align the Mel-frequency cepstral coefficient (MFCC) representation of the two audio sources. We use forty-dimensional MFCCs, and to facilitate the transfer [...]”

(3) Can you provide some comparative results?

As we mentioned in the response to Reviewer 1, we cannot provide directly comparable results to other referenced papers due to differences in the data (different datasets [4, 6, 7, 15], different splits and metrics [5]) and missing or incomplete official implementations of others [a, b, c].

For your reference this is the response that we sent to Reviewer 1:

There is little prior work on the full text-to-lip task. The work of Kumar et al. [3] is one such example, but unfortunately their paper does not provide any (quantitative) results and it does not point to an official implementation. While unofficial repositories do exist [a, b], they are incomplete, missing the text-to-speech component. In terms of the speech-to-lip component, it is difficult to compare directly to previous works since these evaluate on different datasets [4, 6, 7, 15] or using different metrics or splits [5]; we have also had difficulties in running the available codebases for some of these methods (for example, the code [c] corresponding to [5] provides only a partial implementation of the method discussed in the paper).

Additional references

[a] https://github.com/karanvivekbhargava/obamanet

[b] https://github.com/acvictor/Obama-Lip-Sync

[c] https://github.com/supasorn/synthesizing_obama_network_training

Best regards,           

Dan Oneață,             

Beáta LÅ‘rincz,             

Adriana Stan,             

Horia Cucu

Reviewer 3 Report

The authors proposed a transformer architecture to convert text to lips keypoints aiming for synthectic media applications. They designed the text-to-keypoints pipeline to allow for objective
evaluation and performed the evaluation of each component independently and at the
system-level, as well. They claim that their work is the first one to evaluate objectively the output of the text-to-keypoints task and that they objectively assess the quality for the text-to-keypoints seen as a whole is one of the important contributions of their work. The approach is based on two independent modules: a text-to-speech synthesis system and a speech-to-lip one.
For the text-to-speech synthesis component (TTS) they selected one of the latest deep neural-based architectures, able to generate speech which is very close to the natural one called FastPitch [18].

They proposed a Transformer which has two main components: an encoder module that uses self-attention layers to pool the input audio, and a decoder module that uses attention layers to aggregate information from both the encoded audio and previously generated lips. The decoder predicts the lips keypoints at each time step in an autoregressive manner, and is exposed to the entire input audio sequence.

The approach requires a training of the decoder because the speech recognition decoder is designed to output a sequence of characters, while in the task of the approach output is a sequence of lips keypoints. They used videos and transcripts of Obama and Trump from YouTube in order to create their database to train and test the performance. The achieved results are presented both as objetive measures as with demonstrations with texts, audios and lips visualizations in their site  https://zevo-tech.com/humans/flexlip/.

The contributions are very interesting and the experiments are convincing. Although the explanations are satisfactory I would suggest that the authors include few diagrams to help to explain the architectures employed and proposed modifications.

Author Response

26th of May, 2022

Response letter for the review of MDPI sensors-1750936

Dan Oneață, Beáta LÅ‘rincz, Adriana Stan, Horia Cucu,

”FlexLip: A controllable text-to-lip system”

We would like to first thank our editor and reviewers for the time and effort they have put in to evaluate our work, and for the extremely useful comments and suggestions they made in their reviews. 

We attempted to address all points and listed below our responses:

Reviewer 3:

The contributions are very interesting and the experiments are convincing. Although the explanations are satisfactory I would suggest that the authors include few diagrams to help to explain the architectures employed and proposed modifications.

We provided a schematic overview of our system in Figure 1. We are unsure what other diagrams we should have included. 

Best regards,           

Dan Oneață,             

Beáta LÅ‘rincz,             

Adriana Stan,             

Horia Cucu